# Can self-efficacy mediate between knowledge of policy, school support and teacher attitudes towards inclusive education?

Shirli Werner [1] *, Tom P. Gumpel [2], Judah Koller [2], Vered Wiesenthal [2], Naomi Weintraub [3]

1 Paul Baerwald School of Social Work and Social Welfare, Hebrew University of Jerusalem, Jerusalem, Israel, 2 Seymour Fox School of Education, Hebrew University of Jerusalem, Jerusalem, Israel, 3 Faculty of Medicine, School of Occupational Therapy, Hebrew University of Jerusalem, Jerusalem, Israel

* Shirli.werner@mail.huji.ac.il

**Data Availability Statement:** All data files can be found from the OSF databased at https://osf.io/786an/.

## Abstract

### Background

While research has focused on understanding teachers' attitudes towards the inclusion of children with special educational needs into general education classrooms, there are lacunae that have yet to be addressed. This study examined the association between perceived self-efficacy and attitudes towards inclusion among elementary school teachers. The study also examined the role of teachers' self-efficacy as a mediating variable between knowledge of inclusion policy, perception of school support and teachers' attitudes towards inclusion.

### Methods

Teachers (N = 352) working in general or special education schools completed questionnaires assessing attitudes towards inclusion, sense of self-efficacy, knowledge of current policy, and perception of support for inclusive practices.

### Results

Higher perceived knowledge of inclusion policy and higher perceived school support of inclusion were both related to higher self-efficacy regarding inclusion, which, in turn, was related to more positive attitudes about inclusion.

### Conclusion

Our results suggest that point to being knowledgeable regarding local and national policy is important in order to increase feelings of self-efficacy regarding the implementation of effective educational practice. To enhance inclusion, local and national policy must be clearly communicated to teachers. Furthermore, leadership and a supportive school environment are conducive to successful inclusive education.

**Funding:** This study was conducted with the help of a grant provided by the Ministry of Education.

**Competing interests:** The authors have declared that no competing interests exist.

# Introduction

## School inclusion and teachers' attitudes

School inclusion refers to the acceptance of students with disabilities into general education classrooms while responding to their diverse needs and providing the intervention and support necessary for them to succeed [1]. In Israel, as in many other countries, inclusion is frequently confused with integration or mainstreaming. For the purpose of this article, inclusion refers to the modification and preparation of the school system in order to accommodate for the needs of children with disabilities [2]. While multiple variables are associated with the success of inclusion, teachers' attitudes towards inclusion, which is the focus of this article, are the key determinant [3].

Special education in Israel is guided by the 1988 Special Education Law [4]. This law has led to some, but not sufficient, movement into a more inclusive general education system [5, 6]. Thus, following years of frustration, the Ministry of Education created a blue-ribbon committee panel, the Dorner Committee (as is the practice in Israel, governmental committees are named after the committee chair, in this case a retired Supreme Court Judge, Dalia Dorner), to examine the status of the special education system in Israel [7]. The committee examined the status of inclusionary practices in Israel and made policy recommendations in three primary realms: (1) parental choice of the preferred educational setting for their child (in general or special education settings); (2) budgetary modifications and priorities to encourage inclusion; and (3) focus on an educational model of service provision where placement and treatment modalities are primarily based on the child's functional levels [7]. Albeit this policy change seemed encouraging, inclusionary practices were still only partially implemented, largely because of consistent budgetary problems. While between the years of 2000 to 2018 budgets of the educational system increased, scant change occurred in the number of students included in the inclusive education system, ranging between 38 and 44 percent of students with special educational needs within the general education system [8].

Just as inclusion is a multi-faceted construct, attitudes regarding inclusionary practices are complex. Mahat [3] suggested that attitudes include cognitive, affective, and behavioral dimensions. The cognitive dimension reflects teachers' beliefs regarding inclusion, the affective dimension represents associated emotions, and the behavioral dimension reflects teachers' intention to act in a certain manner. Cullen et al. [9] added an additional perspective to this theoretical framework, suggesting that attitudes are comprised of teachers' perceptions of students with disabilities within inclusive settings, beliefs about the efficacy of inclusion and perceptions of the teachers' own professional roles, functions, and abilities [9].

While teachers' attitudes towards inclusion have been examined in many studies and countries, conflicting findings indicate the need for continued research. Studies show that teachers acknowledge the importance of inclusion [10–13]. For example, pre-service teachers in Australia held generally positive attitudes towards inclusion and these attitudes were associated with years of training [11, 14]. Other studies show that some teachers hold negative attitudes toward inclusion [15], see the inclusive process as problematic [16] and favor a separate special education system [17]. Further, some teachers prefer to not be involved in inclusion [18] and perceive it to be too time consuming [19]. In Israel, teachers felt that inclusion, especially of students with intellectual disabilities, failed due to insufficient resources provided to teachers and the limited professional abilities of general education teachers [18].

As noted above by the percentage of students with special educational needs who are included in the general education system in Israel, it seems that frequently these students are *a priori* initially educated in special education settings. The process of inclusion requires that teachers in special education schools make recommendations regarding the transition of these

students into the general education system, while general education teachers are the main agents in implementing the inclusion policy [20]. Therefore, the attitudes of teachers in both systems are important. Studies comparing the attitudes of teachers in general and special education systems are scarce. Most studies were carried out in general education settings and compared teachers according to their role, finding more positive attitudes among special education versus general education teachers [20, 21].

## Teachers' self-efficacy, knowledge of policy, school support and attitudes towards inclusion

Different factors are associated with teachers' attitudes towards inclusion. One frequently examined factor is teachers' self-efficacy beliefs. Self-efficacy beliefs are future-oriented beliefs relating to the individuals' confidence and perceived ability to perform a given behavior [22]. Teachers' self-efficacy beliefs relate to their confidence and perceived ability to provide academic instruction, create a positive learning environment, and influence how well their students learn [23, 24]. Teachers with low self-efficacy beliefs were more likely to ascribe difficulties in learning to the child and less willing to adapt their teaching methods [25].

The literature examining the association between teacher self-efficacy beliefs and their attitudes towards inclusion is inconclusive. Some found a relationship between teacher' self-efficacy belief and successful inclusive teaching [24] or their attitudes toward inclusion [26–28]. Others found a weak relationship between these variables [10, 17]. Still others indicate that only certain aspects of self-efficacy belief, such as efficacy in collaboration with other teachers and parents [29] or regarding instructional strategies [24], were related to attitudes towards inclusion.

Beyond self-efficacy beliefs, two additional variables are important in understanding attitudes towards inclusion: knowledge of inclusion policy and school support of inclusion. We contend that these two variables may have a direct association with attitudes towards inclusion, as well as an indirect association via their possible connection with self-efficacy beliefs. Previous studies that examined the association between knowledge, school support and inclusion have largely focused on inclusionary practices or implementation of inclusion as opposed to attitudes towards inclusion [e.g. 30, 31]. Since attitudes towards inclusion are a key determinant of inclusive practice [3], we maintain that the association found regarding inclusion implementation is also relevant to attitudes toward inclusion.

Teacher's knowledge regarding inclusion policy was selected for this study given that policy is at the heart of understanding the historical narrative and development of the special education field [32]. Examination of local and national policy and knowledge of this policy provides information on the perspectives towards equality and equal rights of participation of children with disabilities within a specific educational system [33]. Further, teacher's knowledge regarding praxis, skills and implementation of inclusion is an important predictor of successful implementation of inclusion [e.g. 30, 31, 34]. As Mieghem and colleagues show in a recent systematic review [35] attitudes towards inclusive education are influenced by teacher's knowledge of disabilities and their experience with inclusive education. Studies have also shown knowledge to be related to self-efficacy beliefs given that teachers who are knowledgeable feel they have skills and confidence in implementing inclusionary practices [21, 36].

In the current study, we focus on knowledge regarding inclusion policy. Examining knowledge regarding inclusion policy is significant because in order for teachers to successfully implement inclusion in their school system, they must be familiar with relevant local and national policy. This is especially important in Israel, a country in which school inclusion has

progressed slowly due to competing socio-historical trends favoring segregation [37]. Thus, understanding of such policy is likely related to attitudes towards implementation of that policy. Further, examining this factor is in line with Weatherley and Lipsky's notion of "street-level bureaucrats" [38, p. 172], seeing the lower-level staff as being central in the policy-making process and suggesting that the meaning of policy must be clear and transparent at the street level. Thus, we examined the association between teachers' knowledge of inclusion policy and their attitudes towards inclusion as well as how this relationship may be mediated by teachers' perceived self-efficacy beliefs.

School support of inclusion was selected for this study based on studies stressing the vital importance of school leadership and receiving support within the educational system in which they work [39]. School leadership has been consistently linked to teacher self-efficacy beliefs and attitudes [40–42] and has a direct relationship with school organizational climate [43, 44] and institutional support [24]. Further, previous studies show that school climate, support from school leadership (e.g., administrators), supportive relationships among teachers and the attitudes of other teachers are important factors in successful inclusion [24, 45]. Beyond its association with attitudes towards inclusion, the principal's leadership and relationships between teachers appear to be related to teacher's professional self-efficacy. Teachers who believe they work in a positive and supportive school environment are more likely to perceive themselves as capable of teaching students with disabilities [24]. Thus, teachers' self-efficacy beliefs may mediate between school support of inclusion and attitudes towards inclusion.

Put together, teacher beliefs in self-efficacy in implementing inclusionary practices have been found to be related to their attitudes regarding inclusion [46, 47]. Likewise, policy engagement and understanding is a clear aspect of organizational health and has been linked with leader self-efficacy [48]. Further, direct pre- and in-service training is clearly linked to perceived self-efficacy [49].

### Demographic and occupational predictors

Several demographic and occupational variables appear relevant to understanding teachers' attitudes towards inclusion. Some research shows that female teachers typically hold more positive attitudes towards inclusion than males [10, 14, 17, 50, 51], while others found no gender-based differences [52, 53]. Some research points to teachers' age being unrelated to attitudes towards inclusion [54], while other studies show that younger teachers have more positive attitudes [10, 14, 51, 55].

Regarding training and experience, again, the literature paints a conflicting picture. Some studies report that higher levels of education and fewer years of teaching experience were associated with more positive attitudes towards inclusion [26, 56]. In contrast, Yadav et al. [53] found that teachers with fewer than ten years of experience expressed most concerns about inclusion. One consistent finding points to the fact that training in special and inclusive education is correlated with positive attitudes towards inclusion [51, 56–59].

### Aims of the current study

While much attention has been paid to understanding teachers' attitudes towards inclusion, there are relative lacunae that have yet to be addressed. First, while some studies compared attitudes towards inclusion between general and special education teachers, few studies, and none conducted in Israel, have compared these attitudes in reference to teachers' work setting (general versus special education schools). Second, while some studies have examined the role of knowledge regarding inclusion practice, they did not specifically focus on knowledge of relevant policy. Third, we could find no studies examining the mediating role of teachers'

professional self-efficacy in implementing inclusionary practices, in the relationship between knowledge of inclusionary policy, perception of school support and attitudes towards inclusion.

The aims of the current study are therefore threefold. First, to examine the association between self-efficacy and attitudes towards inclusion among Israeli elementary school teachers who worked in general and special education schools. Second, to examine differences in attitudes towards inclusion between teachers working in general versus special education settings. Third, to examine the role of general education teachers' self-efficacy beliefs as a mediating variable between knowledge of inclusion policy, perception of school support and teacher attitudes towards inclusion.

Specifically, we examined the following hypotheses:

1. Associations will be found between background (demographic, educational, and occupational) variables and attitudes toward inclusion. Teachers with a higher academic degree, fewer years of experience, homeroom teachers, special education teachers, and those with more experience in teaching students with special educational needs will report more positive attitudes towards inclusion. (In the Israeli educational system, primary and secondary students have several content-based teachers (i.e., math or science); however, all students have one primary teacher (the so-called "homeroom" teacher) who has primary educational responsibility for all of his or her students and acts as an "school-based case manager.")

2. Teachers from general education schools will report increased self-efficacy and more positive attitudes toward inclusion compared to teachers from special education schools.

3. Among teachers from general education schools, a positive association will be found between knowledge of inclusion policy, school support and self-efficacy as well as attitudes towards inclusion.

4. Among teachers from general education schools, self-efficacy beliefs will mediate the relationship between knowledge of inclusion policy, school support, and attitudes toward inclusion.

## Materials and methods

### Participants

Participants were 352 teachers, mostly female (91.6%), who worked either in general (n = 252) or special (n = 100) education schools. More respondents (58%) had training in general education and 42% were special educators. Teachers' mean age was 41.5 (SD = 10.48); 61.0% reported having either a BA or BEd degree and 39.0% reported having an MA degree. Teachers had a mean of 15.0 years of experience (SD = 10.15), and 85.8% had taught students with special educational needs in the past. Only 21.6% of teachers reported receiving some type of training regarding inclusion. Finally, 60.6% were homeroom teachers and 39.4% were content area teachers.

### Instruments

Teachers completed the following measures:

*Attitudes towards inclusion* were measured using two scales, in an attempt to capture a relatively comprehensive and more conceptually complete picture of teachers' attitudes towards inclusion. All scale items selected were relevant to the Israeli context and, thus, no cultural-adaptation was needed. Eighteen items were taken from the Multidimensional Attitudes

Toward Inclusive Education Scale [MATIES; 3]. For example, "I believe an inclusive school is one that permits academic progress of all students regardless of their ability." Items were rated on a 5-point scale ranging from 1 ("disagree very strongly") to 5 ("very strongly agree"). The scale has three factors (each having 6 items): cognitive, affective and behavioral aspects of attitudes. Internal consistency was reported as between $\alpha = .77$ to $\alpha = .91$ in previous studies [3], and $\alpha = .76$, $\alpha = .81$, and $\alpha = .89$ in the current study. We calculated a mean score for each of the three factors by averaging its items.

Twelve items were taken from the Teacher's Attitudes Toward Inclusion Scale [TATIS; 9]. We removed two items from the original scale, as they overlapped with items in the MATIES. In all items, the term "mild to moderate disability" was changed to "moderate to severe disabilities" as we suspected that it would be easier for teachers to report on positive attitudes when it comes to students with less complex disabilities. For example, "It is seldom necessary to remove students with moderate to severe disabilities from regular classrooms in order to meet their educational needs." Items were rated on a 5-point scale ranging from 1 ("disagree") very strongly to 5 ("very strongly agree"). The scale has three factors based on mean scores: Teacher perception of students with moderate to severe disabilities in inclusive settings (4 items, $\alpha = .80$), beliefs about the efficacy of inclusion (4 items, $\alpha = .86$), and perceptions of professional roles and functions (4 items, $\alpha = .68$). Reliabilities in the current study were $\alpha = .76$, $\alpha = .74$ and $\alpha = .70$ for each of the subscales, respectively.

*Teacher's sense of self-efficacy* was measured using the 12-item Teacher's Sense of Efficacy Scale [TSES, 60], which measures teacher evaluations of their own likely success in teaching children with disabilities. Participants were asked to rate to what extent they are able to achieve each of the item statements. For example, "How much are you able to control disruptive behavior in the classroom?" Items were rated on a 5-point scale ranging from 1 ("not at all") to 5 ("to a great extent"). The scale has three factors as well as an overall score. We used the overall score based on its high reliability in previous studies [$\alpha = .90$; 40] and in the current study ($\alpha = .91$).

*Subjective knowledge about inclusion policy* was measured by one item developed for this study: "To what extent do you feel you have knowledge on the principles of the current Ministry of Education's policy about inclusion?" ranked on a 5-point scale from 1 ("very little") to 5 ("very much").

**Perception of school support for inclusion.** Teachers in general education schools were asked to answer four items developed specifically for this study. Items were developed from our discussions with teacher and general and special education administrators: To what extent do you feel that: (1) you receive support from your school in dealing with students who are included? (2) the educational staff in your school is minded towards inclusion? (3) your school principal is focused on inclusion?, and (4) you are satisfied from the inclusion program in their school? Each item was rated on a 5-point scale ranging from 1 ("very little") to 5 ("very much"). Factor analysis resulted in a one factor solution, with all four items having a loading in the range of 0.81 to 0.85 explaining 69.78% of the variance. A mean score was calculated for all four items with reliability of $\alpha = .85$.

**Demographic and work-related variables.** These included gender, age, teacher role (homeroom or content area teacher), academic degree, years of teaching experience, training (general or special education), experience in working with students with special educational needs (yes or no), training in inclusion (binary), type of school (binary: general or special education).

## Procedures

The study protocol was approved by the Hebrew University's School of Social Work IRB. After 2009, the policy recommendations were piloted in 149 schools across four cities in Israel. An

additional pilot was conducted during the 2016–17 academic year. Data for the current study came a random sampling of 43 schools, drawn from the 61 schools that participated in this pilot implementation of the aforementioned inclusion policy, in advance of a national dissemination of said policy. All teachers within the sampled schools were asked to complete the questionnaires; however, in adherence to ethical guidelines, they were not mandated to respond. Most questionnaires were completed by the teachers when a research assistant visited the school, while others completed the questionnaires online. Data for this study were collected as part of a Ministry of Education-sponsored study of educational reforms led by the research team.

## Analyses

Means and standard deviations were used to examine the central tendency of the study variables. For hypothesis 1, Pearson correlations and *t*-tests were calculated to assess the relationships between teachers' self-efficacy and attitudes toward inclusion and the background variables. For hypothesis 2, multivariate analyses of covariance were calculated to examine the differences in teacher self-efficacy and attitudes toward inclusion, between teachers in general education and teachers in special education. For hypothesis 3, multiple hierarchical regressions were calculated for teachers within the general education system, to assess the association between background variables, knowledge, and school support of inclusion and teachers' self-efficacy, and teachers' attitudes toward inclusion. For hypothesis 4, the PROCESS procedure [61] was used to assess the mediating role of teachers' self-efficacy between knowledge about inclusion and school support of inclusion, and teachers' attitudes toward inclusion. Prior to data analysis, we analyzed missing data based on Little's test of Missing Completely at Random (MCAR). Results were non-significant, meaning that there missing data were random and so pairwise deletion was appropriate.

# Results

## Descriptive statistics

Table 1 presents the means and standard deviations of the components of teachers' attitudes towards inclusion and teachers' self-efficacy for the overall sample. As indicated, relatively strong positive attitudes were reported on the behavioral scale and for perception of teachers' professional roles and functions, while moderately positive attitudes were expressed on the

**Table 1. Means and standard deviations of main study variables (N = 352).**

|  | Range | Mean | SD |
|---|---|---|---|
| **Attitudes towards inclusive education** |  |  |  |
| MATIES |  |  |  |
| Cognitive | 1.33–5.00 | 3.39 | 0.76 |
| Affective | 1.00–5.00 | 3.57 | 0.85 |
| Behavioral | 1.00–5.00 | 3.91 | 0.78 |
| TATIS |  |  |  |
| Perception of students with disabilities | 1.00–5.00 | 2.25 | 0.83 |
| Beliefs about efficacy of inclusion | 1.00–5.00 | 2.64 | 0.86 |
| Perception of professional roles and functions | 1.00–5.00 | 3.78 | 0.75 |
| **Teachers' self-efficacy** | 2.00–5.00 | 4.12 | 0.53 |

Note: All scales range from 1 to 5 with higher ratings representing more positive attitudes.

affective and cognitive scales. Less positive attitudes were found regarding teachers' beliefs about the efficacy of inclusion and attitudes below the midpoint of the scale were reported for teachers' perception of students with disabilities within inclusive settings. Overall, most teachers reported high levels of self-efficacy.

## Associations between background variables and attitudes toward inclusion

We first examined the relationship between demographic and occupational variables and dimensions of teachers' attitudes toward inclusion and self-efficacy (hypothesis 1). Significance levels was set at .007 based on Bonferroni corrections for seven comparisons for each background variable.

Results showed that compared to teachers who were not trained to work with children with special education needs, teachers who were trained were more positive in the TATIS subscale perception of students with moderate to severe disabilities ($M = 2.66$, $SD = 0.81$ vs. $M = 2.14$, $SD = 0.80$; $t(341) = 4.89$, $p < .001$, $d = .53$); more positive in the MATIES cognitive component of attitudes ($M = 3.69$, $SD = 0.70$ vs. $M = 3.31$, $SD = 0.75$; $t(341) = 3.82$, $p < .001$, $d = .41$); and more positive in the MATIES behavioral component of attitudes ($M = 4.13$, $SD = 0.75$ vs. $M = 3.84$, $SD = 0.78$; $t(341) = 2.79$, $p = .006$, $d = .30$). Years of teaching experience was negatively correlated with the MATIES affective component of attitudes towards inclusion ($r = -.15$, $p = .005$), suggesting that the higher the teaching experience, the lower the affective aspect of attitude toward inclusion. Teacher self-efficacy beliefs were moderately higher for homeroom teachers ($M = 4.19$, $SD = 0.48$) than for content area teachers ($M = 4.00$ $SD = 0.59$; $t(245.08) = 3.11$, $p = .002$, $d = .40$); and slightly higher for teachers trained to work with children with special educational needs ($M = 4.28$, $SD = 0.44$) than for teachers without such training ($M = 4.06$, $SD = 0.54$; $t(341) = 3.34$, $p < .001$, $d = .36$).

Next, we examined the relationship between demographic and occupational variables and dimensions of teachers' attitudes toward inclusion and self-efficacy only among teachers who worked in general education settings. Compared to teachers who were not trained to work with children with special educational needs, teachers who were trained to work with children with special educational needs reported more positive TATIS perception of students with disabilities ($M = 2.68$, $SD = 0.83$ vs. $M = 2.24$, $SD = 0.83$; $t(243) = 3.81$, $p < .001$, $d = .49$) and higher on the MATIES cognitive component of attitudes ($M = 3.71$, $SD = 0.70$ vs. $M = 3.47$, $SD = 0.67$; $t(243) = 2.47$, $p = .014$, $d = .32$). Years of teaching experience was negatively correlated with the MATIES affective component of attitudes ($r = -.19$, $p = .003$). Further, among teachers who worked in general education settings, reported self-efficacy was moderately higher for teachers trained to work with children with special educational needs ($M = 4.29$, $SD = 0.44$) compared with those without such training ($M = 4.06$, $SD = 0.52$; $t(243) = 3.25$, $p = .001$, $d = .42$). Other demographic and background variables were unrelated with teacher self-efficacy or attitudes toward inclusion. Thus, subsequent analyses controlled for teacher's role, teacher training, and years of teaching experience.

## Differences between teachers in general and special education schools

In Israel, teachers can choose to work in either special or general education schools regardless of the specific training that they hold (this is despite the fact that teachers are licensed for work either in the general or special educational systems). Thus, attitudes toward inclusion and teachers' self-efficacy were compared between teachers in general education and special education settings (hypothesis 2). Multivariate one-way analyses of covariance for the three TATIS subscales (teacher perceptions, beliefs about the efficacy of inclusion, and perceptions regarding professional roles) by school type, with teacher's role (homeroom teacher = 1 vs.

**Table 2. Means, standard deviations and *F* values for teachers' attitudes toward inclusion and self-efficacy (*N* = 332).**

| | General education *M* (*SD*) (n = 235) | Special education *M* (*SD*) (n = 97) | *F*(1, 327) (η²) |
|---|---|---|---|
| TATIS | | | |
| Perception of students with moderate to severe disabilities in inclusive settings | 2.34 (0.84) | 1.97 (0.71) | 7.58** (.02) |
| Beliefs about the efficacy of inclusion | 2.74 (0.84) | 2.31 (0.80) | 14.75*** (.04) |
| Perceptions of professional roles and functions | 3.80 (0.76) | 3.72 (0.73) | 0.28 (.00) |
| MATIES | | | |
| Cognitive | 3.52 (0.68) | 3.03 (0.81) | 21.80*** (.06) |
| Affective | 3.48 (0.83) | 3.81 (0.86) | 9.59** (.03) |
| Behavioral | 3.94 (0.75) | 3.78 (0.85) | 1.75 (.01) |
| Teachers' self-efficacy | 4.13 (0.51) | 4.06 (0.58) | 0.12 (.00) |

**p < .01

***p < .001 Note: All scales range from 1 to 5.

professional teacher = 0), teacher training (trained to work with children with special educational needs = 1 vs. not trained = 0), and years of teaching experience as covariates.

The analysis revealed a main effect for the three TATIS scales, Wilks $\Lambda$ = 0.95, F $_{(3, 325)}$ = 6.10, $p < .001$, η² = .05. Among the covariates, only teacher training [Wilks $\Lambda$ = 0.97, F $_{(3, 325)}$ = 3.93, $p < .01$, η² = .04] and teaching seniority [Wilks $\Lambda$ = 0.97, F $_{(3, 325)}$ = 3.12, $p < .05$, η² = .03] were significant. Follow-up tests of between subject effects only for school type, showed differences, but with small to moderate effect sizes, only for the teacher perception scale [F $_{(1, 327)}$ = 7.56, $p < .05$, $η_p^2$ = .02] and the teacher belief scale [F $_{(1, 327)}$ = 14.76, $p < .001$, $η_p^2$ = .04].

A similar MANCOVA analysis was performed based on the three MATIES scales (cognitive, affective, and behavioral) using the same covariates. The omnibus test revealed a main effect for school type [Wilks $\Lambda$ = 0.84, F $_{(3, 325)}$, $p < .001$, η² = .16] along with significant main effects for the teacher training [Wilks $\Lambda$ = 0.98, F $_{(3, 325)}$, $p < .05$, η² = .03] and teaching seniority [Wilks $\Lambda$ = 0.96, F $_{(3, 325)}$ = 4.70, $p < .01$, η² = .04] covariates. Follow-up analyses showed significant differences only for the cognitive [F $_{(1, 327)}$ = 21.80, $p < .001$, $η_p^2$ = .06] and the affective subscales [F $_{(1, 327)}$ = 9.59, $p < .01$, $η_p^2$ = .03].

Table 2 shows that perception of students with moderate to severe disabilities in inclusive settings, along with beliefs about the efficacy of inclusion were more positive among teachers in general education than among teachers in special education settings. Cognitive component of attitudes toward inclusion were more positive among teachers in general education than in special education, while the opposite trend is evident concerning the affective component of attitudes.

## Relationships between background variables, subjective knowledge of inclusion policy, school support of inclusion and teachers' self-efficacy and attitudes toward inclusion

As indicated above, knowledge of inclusion policy and perception of school support were examined only among teachers working in the general education system. Mean subjective knowledge for these teachers was 2.21 (SD = 1.10) and mean school support of inclusion was 3.76 (SD = 0.88). Two sets of multiple hierarchical regressions were calculated. In the first set, teacher self-efficacy was examined as the dependent variable. In the second set, teachers'

attitudes towards inclusion was the dependent variable with each attitude factor examined separately. For the regression predicting self-efficacy, independent variables were knowledge of inclusion policy and school support. For the regression predicting each attitude factor, independent variables were knowledge of inclusion policy, school support, and teachers' self-efficacy. As before, we controlled for the teacher's role (homeroom teacher = 1 vs. professional teacher = 0), teacher training (trained to work with children with special needs = 1 vs. not trained = 0), and years of teaching experience. In all regressions, control variables were entered in the first step, knowledge of inclusion policy and school support of inclusion were entered in the second step and teachers' self-efficacy was entered in the third step (Table 3).

Results show that all regression models were significant, although rates of explained variance were quite low. Teachers' self-efficacy was predicted by their training, knowledge of inclusion policy, and school support. Unsurprisingly, teachers who were trained to work with children with special educational needs, had more knowledge of inclusion policy, and who perceived their schools to be more supportive of inclusion, reported higher levels of self-efficacy regarding inclusion.

Interestingly, teachers' perceptions of students with moderate to severe disabilities in inclusive settings were predicted by training and school support, such that teachers who were

**Table 3. Multiple regressions for teacher self-efficacy and attitudes toward inclusion in the general education system ($N = 235$).**

| | | TATIS | | | MATIES | | |
|---|---|---|---|---|---|---|---|
| | Teacher self- efficacy | Perc. of students | Beliefs | Perc. of role and function | Cognitive | Affective | Behavioral |
| | B | β | β | β | β | β | B |
| **Step 1** | | | | | | | |
| Teachers' role[1] | .15* | -.04 | -.03 | .05 | -.01 | .03 | -.07 |
| Training | .21** | .20*** | .05 | .08 | .16* | .04 | .19** |
| Years of Experience | -.02 | .10 | .10 | -.12 | -.02 | -.17** | -.10 |
| Adj.$R^2$ | .05** | .04** | .001 | .01 | .01 | .02 | .03* |
| **Step 2** | | | | | | | |
| Teachers' role[1] | .09 | -.08 | -.06 | .03 | -.05 | .01 | -.11 |
| Training | .14* | .14* | -.01 | .05 | .11 | -.02 | .13* |
| Years of Experience | -.02 | .11 | -.08 | -.11 | -.01 | -.16* | -.08 |
| Knowledge | .28*** | .14* | .04 | .07 | .07 | .07 | .08 |
| School support | .17** | .24*** | .32*** | .11 | .28*** | .27*** | .31*** |
| ΔAdj.$R^2$ | .11*** | .08*** | .09*** | .01 | .08*** | .07*** | .11*** |
| **Step 3** | | | | | | | |
| Teachers' role[1] | -- | -.09 | -.07 | .01 | -.07 | -.02 | -.11 |
| Training | -- | .13* | -.02 | .02 | .08 | -.05 | .09 |
| Years of Experience | -- | .11 | -.08 | -.11 | -.01 | -.16** | -.08 |
| Knowledge | -- | .11 | .02 | .01 | .02 | .01 | .01 |
| School support | -- | .22*** | .30*** | .07 | .25*** | .23*** | .27*** |
| Self-efficacy | -- | .10 | .08 | .23** | .21** | .23*** | .27*** |
| ΔAdj.$R^2$ | -- | .01 | .001 | .04** | .04** | .04*** | .05*** |
| Adj.$R^2$ | .16 | .13 | .09 | .06 | .13 | .13 | .19 |
| $F$ (6, 228) | 10.23*** | 6.65*** | 5.07*** | 3.43** | 6.64*** | 6.90*** | 10.20*** |

*$p < .05$

**$p < .01$

***$p < .001$ For self-efficacy df = 5, 229.

[1] Teachers' role = homeroom teacher versus professional teacher.

trained to work with children with special educational needs, and who perceived their schools to be more supportive of inclusion, reported more positive perceptions of students with disabilities in inclusive settings.

Teachers' beliefs about the efficacy of inclusion were explained only by school support. In other words, teachers who perceived their schools to be more supportive of inclusion reported more positive beliefs about its efficacy. Teachers' perceptions of professional roles and functions were significantly explained only by their self-efficacy in the context of inclusion, so that teachers with high self-efficacy reported more positive perceptions of professional roles and functions regarding inclusion.

Cognitive, affective, and behavioral attitudes toward inclusion were primarily explained by school support of inclusion and teacher self-efficacy regarding their abilities to engage in inclusionary practices. Teachers who perceived their schools to be more supportive of inclusion, and who had higher self-efficacy, reported more positive cognitive, affective, and behavioral attitudes toward inclusion. In addition, teachers with less experience in teaching reported more positive affective attitudes toward inclusion (hypothesis 3).

These results reveal that teacher self-efficacy regarding inclusion may mediate the relationships between knowledge of inclusion policy and school support, and several of the attitudes towards inclusion factors (namely, teachers' perceptions of professional roles and functions, cognitive, affective, and behavioral attitudes). These possible mediations, described in hypothesis 4, were examined using the PROCESS analysis [61]. As before, the teacher role and teacher training were controlled for.

Results presented in Table 4 show that all mediation models were significant, as all confidence intervals for the indirect effects were positive and all Z values were significant. Further,

**Table 4. Path coefficients and indirect effects for self-efficacy as a mediator between knowledge and school support and teachers' attitudes toward inclusion** ($N = 235$).

| Dependent Variable (DV) | Variable | Path Coefficients | | Indirect effects | | |
|---|---|---|---|---|---|---|
| | | to DV Estimate (SE) | to Mediator Estimate (SE) | Estimate (SE) | Z (d) | 95% CI |
| **TATIS** | | | | | | |
| Perc. of role and function | Knowledge | .013 (.054) | .312 (.064)*** | .057 (.018) | 3.103** (0.413) | .024, .095 |
| | Self-efficacy | .184 (.053)*** | | | | |
| Perc. of role and function | School support of inclusion | .055 (.049) | .210 (.062)*** | .037 (.015) | 2.480* (0.328) | .011, .068 |
| | Self-efficacy | .175 (.051)*** | | | | |
| **MATIES** | | | | | | |
| Cognitive attitudes | Knowledge | .029 (.047) | .312 (.064)*** | .054 (.017) | 3.158** (0.421) | .023, .091 |
| | Self-efficacy | .171 (.047)*** | | | | |
| Cognitive attitudes | School support of inclusion | .180 (.042)*** | .210 (.062)*** | .029 (.012) | 2.387* (0.315) | .008, .056 |
| | Self-efficacy | .141 (.044)** | | | | |
| Affective attitudes | Knowledge | .022 (.058) | .312 (.064)*** | .071 (.022) | 3.194** (0.426) | .030, .117 |
| | Self-efficacy | .228 (.056)*** | | | | |
| Affective attitudes | School support of inclusion | .176 (.052)*** | .210 (.062)*** | .038 (.016) | 2.311* (0.305) | .010, .075 |
| | Self-efficacy | .181 (.054)*** | | | | |
| Behavioral attitudes | Knowledge | .021 (.049) | .312 (.064)*** | .074 (.021) | 3.526*** (0.473) | .036, .118 |
| | Self-efficacy | .229 (.048)*** | | | | |
| Behavioral attitudes | School support of inclusion | .159 (.046)*** | .210 (.062)*** | .035 (.015) | 2.305* (0.304) | .008, .069 |
| | Self-efficacy | .169 (.048)*** | | | | |

*$p < .05$
**$p < .01$
***$p < .001$.

all mediation models reveal the same pattern. Higher perceived knowledge of inclusion policy was related to higher self-efficacy regarding inclusion, which, in turn, associated with more positive attitudes about inclusion. Similarly, higher perceived school support of inclusion was related to higher self-efficacy regarding inclusion, which was related with more positive attitudes about inclusion.

## Discussion

This study had two broad aims: First, to examine the association between self-efficacy and attitudes towards inclusion among Israeli elementary school teachers in general and special education schools. Second, among teachers working in general education schools, to examine the role of teacher self-efficacy as a mediating variable between knowledge of inclusion policy, perception of school support, and teacher attitudes towards inclusion.

Results of the current study indicate that teachers rate themselves highly on self-efficacy regarding their perceived ability to teach students with disabilities. While this is encouraging, other findings from our study are cause for concern. First, though self-efficacy was associated with training, only 21% of teachers in our study reported receiving any sort of training on inclusion. Second, teachers rated their attitudes towards students with disabilities in inclusive settings and their beliefs about the efficacy of inclusion as relatively low. Third, although their perception of school support regarding inclusion was moderate, teachers perceived their subjective knowledge of the relevant inclusion policy as quite low. Taken together, these findings raise concerns regarding the state of inclusive education in Israel. This will be expanded upon below in discussing the results of our mediation models.

A unique contribution of the current study is its focus on teachers in both special and general education settings. We argue that attitudes of teachers in both these settings are of importance as each plays a role in the success of an inclusive education system. While the attitudes reported by teachers in both these settings were only moderate, those of teachers within the general education system were relatively higher on most scales. These results are somewhat different than those of previous studies that reported on more positive attitudes among special education teachers, albeit most these studies were conducted in general education settings [20, 21].

On the one hand, these findings are encouraging given that teachers who work within general education settings are the main agents in implementing inclusion [20]. On the other hand, these findings are also a cause for concern, since we posit that teachers in special education settings must be aware of the ideological basis and practical significance of inclusion in order to make appropriate recommendations regarding transferring students with special educational needs into general settings. As we have stated elsewhere, there can be no policy without a supportive ideology, and no praxis without supportive policy [2].

### Discussion of mediation models: Knowledge of inclusion policy, school support of inclusion, and self-efficacy

Findings from our mediation models show that while knowledge of inclusion policy was not directly associated with teachers' attitudes towards inclusion, indirect associations were significant via self-efficacy. Specifically, greater familiarity with inclusion policy was associated with greater self-efficacy and the latter was associated with teacher perception of their professional roles and functions, cognitive, affective and behavioral attitudes.

Somewhat differently from previous studies [e.g. 30, 31, 34], our study examined knowledge of inclusion policy rather than knowledge regarding the practice of inclusion itself. Similar to those previous studies, however, our findings highlight the importance of knowledge and our

results point to the importance of being knowledgeable in local and national policy as a vital step towards greater feelings of self-efficacy in being able to implement effective educational practice. This is particularly important since, as suggested by other studies [24] and our own results, teachers with increased self-efficacy, are more likely to have positive attitudes towards inclusion. While our study was cross-sectional and, therefore, does not allow us to examine causation, it is possible that the association is in the opposite direction, i.e. teachers who hold more positive attitudes towards inclusion would report greater self-efficacy.

Based on the current findings, it is clear that in order to enhance inclusion in the Israeli educational system, local and national policy must be clearly communicated to teachers. Teachers must become knowledgeable in the current policy, its ideological basis, and means by which is should be implemented. In keeping with the 3P policy model (Philosophy, Policy, and Praxis) [2], teachers must understand the philosophy and ideology behind the policy before being asked to take an active role in implementing it [2].

Our findings also indicate that school support of inclusion had both a direct and an indirect association with teachers' attitudes towards inclusion. Specifically, higher levels of school support regarding inclusion were associated with greater self-efficacy, while the latter was associated with teachers' perception of their professional roles and functions, as well as cognitive, affective and behavioral attitudes. Greater school support of inclusion was also directly associated with cognitive, affective and behavioral attitude factors.

These findings provide strong support for the importance of leadership within schools [24, 39]. Working in a positive and supportive school environment may provide teachers with enhanced feelings of resilience regarding their capacity to teach students with disabilities [24]. This, in turn, may lead to more positive teacher attitudes towards the inclusion of these children. These findings also highlight the importance of provision of advanced training to teachers at all levels. Specifically, all teachers, including homeroom teachers, content area teachers and special education teachers should be provided with training in inclusive education. This training should be provided beginning with the earliest stages of their careers and should continue on an ongoing basis.

## Study limitations and future research

The strength of the current study is in its examination of teachers in both special and general education settings and in the collection of data from a large sample of schools across the country. One limitation is its focus on attitudes towards inclusion rather than examining the actual practice of teachers working with students with disabilities. An additional limitation was that knowledge about policy of inclusion was measured with a single item. Additionally, this study is based on self-reports which may be susceptible to social-desirability bias. Since our study was anonymous we hope this did not occur. If it did, this would indicate that actual attitudes might be even lower than those reported.

Future studies should focus on actual practice implemented by teachers in various school settings. This could potentially be achieved by self-reports, but would best be measured via observational protocols within class time. Further, studies should focus on the effects of inclusion on students with disabilities, focusing on student academic and social achievements. One possible avenue for conducting such studies is by utilizing student self-report of their experiences in their class and school.

## Conclusions and implications

Findings from this study highlight the importance of knowledge of inclusion policy, school support of inclusion, teacher's self-efficacy, training, and working in general education settings

as being important predictors of teachers' attitudes towards inclusion. In terms of implications regarding the advancement of the field of inclusive education, teacher training programs should not distinguish between the training of special education and general education teachers [2]. This would send a message that all teachers, regardless of their workplace must be capable of teaching heterogeneous classrooms and knowledgeable regarding inclusionary practices.

## Acknowledgments

We would like to thank all research assistants who helped in carrying out this study. Special thanks to Tziva Elgart for her assistance.

## Author Contributions

**Conceptualization:** Shirli Werner, Tom P. Gumpel, Judah Koller, Naomi Weintraub.

**Formal analysis:** Shirli Werner, Tom P. Gumpel, Vered Wiesenthal.

**Funding acquisition:** Shirli Werner, Tom P. Gumpel, Judah Koller, Naomi Weintraub.

**Methodology:** Shirli Werner, Judah Koller, Naomi Weintraub.

**Project administration:** Shirli Werner, Vered Wiesenthal, Naomi Weintraub.

**Supervision:** Shirli Werner, Tom P. Gumpel, Judah Koller, Vered Wiesenthal, Naomi Weintraub.

**Writing – original draft:** Shirli Werner.

**Writing – review & editing:** Tom P. Gumpel, Judah Koller, Vered Wiesenthal, Naomi Weintraub.

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
