## [Decision Letter · Decision Letter 0]

4 Mar 2021

PONE-D-20-30355

Can self-efficacy mediate between knowledge of policy, school support and teachers' attitudes towards inclusive education?

PLOS ONE

Dear Dr. Werner,

Thank you for submitting your manuscript to PLOS ONE. After careful consideration, we feel that this is an interesting topic which deserves further exploration but does not fully meet PLOS ONE’s publication criteria as it currently stands. Therefore, we invite you to submit a revised version of the manuscript that addresses the points raised during the review process. Specifically,  the theoretical grounding of this study needs to be strengthened. The reviewers also raised concerns with other aspects. Please attend to each of them.

We look forward to receiving your revised manuscript.

Kind regards,

Mingming Zhou, Ph.D.

Academic Editor

PLOS ONE

Journal Requirements:

2) We note that you have indicated that data from this study are available upon request. PLOS only allows data to be available upon request if there are legal or ethical restrictions on sharing data publicly. For more information on unacceptable data access restrictions, please see http://journals.plos.org/plosone/s/data-availability#loc-unacceptable-data-access-restrictions.

Reviewers' comments:

Reviewer's Responses to Questions

**Comments to the Author**

1. Is the manuscript technically sound, and do the data support the conclusions?

Reviewer #1: Yes

Reviewer #2: Yes

Reviewer #3: Yes

2. Has the statistical analysis been performed appropriately and rigorously? 

Reviewer #1: Yes

Reviewer #2: Yes

Reviewer #3: Yes

3. Have the authors made all data underlying the findings in their manuscript fully available?

Reviewer #1: Yes

Reviewer #2: Yes

Reviewer #3: Yes

4. Is the manuscript presented in an intelligible fashion and written in standard English?

Reviewer #1: Yes

Reviewer #2: Yes

Reviewer #3: Yes

5. Review Comments to the Author

Reviewer #1: This study is excellently performed and reported. The conclusions made are based on the results and stress important issues.

The importance of school support of inclusion was a nice finding. The authors state that "teachers with increased self-efficacy are more likely to have positive attitudes towards inclusion" While the authors have been generally keen not to mix correlation and causation, they could consider the possibility that causal links here could go in both directions, i.e. positive attitude toward inclusion may itself create more self-sufficiency among teacher (If a person wants something, he probably also thinks: I can manage this). - Howeve, the paper is perfect and does not need additional discussion on this topic.

Reviewer #2: The Authors,

Thank you for giving me the opportunity to read manuscript “Can self-efficacy mediate between knowledge of policy, school support and teachers' attitudes towards inclusive education?”

It is indeed a new learning and insight about inclusive education. I congratulate the authors for taking up specially a less explored variable i.e. teacher’s knowledge about inclusive education policy.

While I found the paper quite relevant in today’s context, I also have some serious concerns in the manuscript as several points are ambiguous. I have highlighted in the manuscript with comments. I also mention them below:

1. The references are mentioned as numbers which makes it difficult to see the reference to make a connect, is this the way the journal asks to give references? why line 138 has a study name? (Yadav et al.) when other studies references are numbered? please be consistent. since I have used most of the references in my work hence, I have a fair idea about the studies cited here.

2. I strongly recommend to strengthen the argument of relation between attitudes and knowledge about country inclusive policy. This can be done by giving specific information about Israel inclusive education policy and the gaps between reaching to “streets” as the authors mention, hence the need to measure it. In the current form it is loosely knit and sounds less convincing as there are lot of studies on attitudes and self efficacy.

3. The link can be: attitudes → self efficacy → knowledge about policy →need for training module . may be this is a part of a bigger study but in the current form it requires a firm argument.

4. I wonder what was the specific reason for picking up Self efficacy and knowledge about policy of education needs. Maybe it has a specific relevance to Israeli context but it has not been brought about. I wonder because there are loads of literature on teacher’s self-efficacy towards inclusion. Perhaps the authors need to make a connection in the specific country context. Moreover, there should be more information about the country specific inclusive education national policy for international readers.

5. The three aspects on which the manuscript focuses are : self efficacy, attitudes , knowledge about policy of inclusive education. Then why other variable such as school leadership , climate support system etc?

6. About the section which has mentioned aims of the study: how are 3a and 3b different? Why have authors decided to have two parts?

7. About the instrument: why were two scales chosen to measure the attitudes? When most of the items are overlapping? The MATIES is quite comprehensive and fits in the theoretical framework also. Only one item to measure knowledge about policy of inclusion? Moreover, is there any specific policy in Israel? what is the significance of teachers to know it? Generally, decisions to make school inclusive is taken by the school leaders or management not the teachers. Teachers play key role in implementing it.

8. About the instrument: details such as final items of the scale should be provided and the last part (see highlighted comments within the manuscript) seems incomplete.

9. The demographic information should be provided in a table which makes it easier to read.

10. Paper requires reorganization and the manuscript has potential to contribute in the research body. It requires a consistency in the use of terms to avoid confusions and make clear for international readers.

11. A suggestion for clear organization is as under:

The three aspects of inclusion in Israel

see the document

Reviewer #3: The study examines an important area of school inclusion. Overall, complex and coherent calculations are made. Minor revisions are needed. The theory foundation should define more precisely the understanding of inclusion (narrow/wide) in the country studied (line 39). Master's theses or doctoral dissertations should be replaced in the literature with more recent international studies (line 41, 52). There are many Finnish studies, partly missing other European and American countries (line 56). The explanation for the country description is missing (line 61)

6. PLOS authors have the option to publish the peer review history of their article (what does this mean?). If published, this will include your full peer review and any attached files.

Reviewer #1: No

Reviewer #2: **Yes: **Meenakshi Srivastava (PhD)

Reviewer #3: No

---

## [Author Response · Author response to Decision Letter 0]

7 May 2021

May 3rd, 2021

Prof. Mingming Zhou, Ph.D.; Academic Editor

RE: PONE-D-20-30355 - Can self-efficacy mediate between knowledge of policy, school support and teachers' attitudes towards inclusive education?

Dear Dr. Zhou,

We would like to thank you and the reviewers for your helpful feedback and comments on this manuscript. We have attempted to revise the manuscript accordingly. Please see our responses below to each comment.

1. The references are mentioned as numbers which makes it difficult to see the reference to make a connect, is this the way the journal asks to give references? since I have used most of the references in my work hence, I have a fair idea about the studies cited here. 

References are numbered according to Plos One guidelines.

2. I strongly recommend to strengthen the argument of relation between attitudes and knowledge about country inclusive policy. This can be done by giving specific information about Israel inclusive education policy and the gaps between reaching to “streets” as the authors mention, hence the need to measure it. In the current form it is loosely knit and sounds less convincing as there are lot of studies on attitudes and self efficacy. 

We have significantly revised the introduction, to which we have added information on Israel’s special education law and changes in policy in this field.

3. The link can be: attitudes → self efficacy → knowledge about policy →need for training module . may be this is a part of a bigger study but in the current form it requires a firm argument. A suggestion for clear organization is as under: 

The four aspects of inclusion in Israel

While this model is intriguing and worthy of future exploration, we cannot speak to its viability based on the current data. The model we propose here reflects the variables examined in the context of this study and our conceptualization of the relationships between them. Nevertheless, we believe that our in-depth revision of the introduction section make the theoretical links between the variables in our study clearer.

4. I wonder what was the specific reason for picking up Self efficacy and knowledge about policy of education needs. Maybe it has a specific relevance to Israeli context but it has not been brought about. I wonder because there are loads of literature on teacher’s self-efficacy towards inclusion. Perhaps the authors need to make a connection in the specific country context. Moreover, there should be more information about the country specific inclusive education national policy for international readers. 

In this revision, we have added information on Israeli’s inclusive education policy. Further, we have attempted to clarify the study model, the associations that examined in this study, and the significance of this work, both in general, and specifically in the Israeli context. 

5. The three aspects on which the manuscript focuses are: self-efficacy, attitudes , knowledge about policy of inclusive education. Then why other variable such as school leadership , climate support system etc? 

In the section titled “Teachers' self-efficacy, knowledge of policy, school support and attitudes towards inclusion”, we clarify that school support of inclusion is one of the variables that we aimed to study. This construct is expressed by school leadership and climate, as defined by previous studies in the field. 

6. About the section which has mentioned aims of the study: how are 3a and 3b different? Why have authors decided to have two parts? 

We have removed hypothesis 3b.

7. About the instrument: why were two scales chosen to measure the attitudes? When most of the items are overlapping? The MATIES is quite comprehensive and fits in the theoretical framework also. Only one item to measure knowledge about policy of inclusion? Moreover, is there any specific policy in Israel? what is the significance of teachers to know it? Generally, decisions to make school inclusive is taken by the school leaders or management not the teachers. Teachers play key role in implementing it. 

We decided to use the two scales as we felt that, in combination, they provided a more comprehensive picture of teachers’ attitudes towards inclusion. We have added this explanation to the methods section. Indeed, only one item was used to measure knowledge of inclusion policy. We have added this as a limitation of the study. As stated above, in the introduction section, we added information on Israel-specific policy. Furthermore, the introduction now makes the case for the importance of conducting this study specifically among teachers.

8. About the instrument: details such as final items of the scale should be provided and the last part (see highlighted comments within the manuscript) seems incomplete. 

Example items from the scales are provided. We did not use an overall scale score. Additional information has been provided to further describe the scales.

9. The demographic information should be provided in a table which makes it easier to read. 

We carefully considered this possibility. While we ordinarily prefer a table presentation, we felt it would be less appropriate for the demographic information in the current paper. The data presented is not complex and includes both continuous and categorical data, such that, we believe a presentation in the text is more straightforward for readers.

10. Paper requires reorganization and the manuscript has potential to contribute in the research body. It requires a consistency in the use of terms to avoid confusions and make clear for international readers. 

Sections of the paper have been significantly revised in response to this comment. We have carefully reviewed the current manuscript to ensure clarity, succinctness, and consistent use of terminology. 

We hope the editor and reviewers will find the revisions made to the manuscript to be satisfactory. We hope this manuscript, in its current format, will be found suitable for publication in Plos One. Thank you again for the opportunity to revise this article.

---

## [Decision Letter · Decision Letter 1]

17 Jun 2021

PONE-D-20-30355R1

Can self-efficacy mediate between knowledge of policy, school support and teachers' attitudes towards inclusive education?

PLOS ONE

Dear Dr. Werner,

Thank you for submitting your manuscript to PLOS ONE. After careful consideration, we feel that it has merit but does not fully meet PLOS ONE’s publication criteria as it currently stands. Therefore, we invite you to submit a revised version of the manuscript that addresses the points raised during the review process.

We look forward to receiving your revised manuscript.

Kind regards,

Mingming Zhou, Ph.D.

Academic Editor

PLOS ONE

Journal Requirements:

Reviewers' comments:

Reviewer's Responses to Questions

**Comments to the Author**

1. If the authors have adequately addressed your comments raised in a previous round of review and you feel that this manuscript is now acceptable for publication, you may indicate that here to bypass the “Comments to the Author” section, enter your conflict of interest statement in the “Confidential to Editor” section, and submit your "Accept" recommendation.

Reviewer #2: (No Response)

Reviewer #3: All comments have been addressed

2. Is the manuscript technically sound, and do the data support the conclusions?

Reviewer #2: Partly

Reviewer #3: Yes

3. Has the statistical analysis been performed appropriately and rigorously? 

Reviewer #2: Yes

Reviewer #3: Yes

4. Have the authors made all data underlying the findings in their manuscript fully available?

Reviewer #2: Yes

Reviewer #3: Yes

5. Is the manuscript presented in an intelligible fashion and written in standard English?

Reviewer #2: Yes

Reviewer #3: Yes

6. Review Comments to the Author

Reviewer #2: Thank you for incorporating the suggestions. the manuscript is more comprehensive now,

however, I still struggle with the link between self efficacy and knowledge about inclusion policy. specifically point 3 AND 5 of the previous comments. Those points still need to be addressed clearly.

Reviewer #3: The article was revised in accordance with the review. Explanations of Israeli policy can be found and the presumed connections have been presented conclusively. Overall, complex and conclusive calculations are made. The theoretical foundation for understanding inclusion in the country under study is now defined. Master's theses or dissertations have been replaced by recent international studies. More country studies have been added. The explanation of the country description is now available. The revision now presents the country-specific regulation and makes clear the importance of the study for Israel.

7. PLOS authors have the option to publish the peer review history of their article (what does this mean?). If published, this will include your full peer review and any attached files.

Reviewer #2: **Yes: **Dr. Meenakshi Srivastava

Reviewer #3: No

---

## [Author Response · Author response to Decision Letter 1]

3 Jul 2021

July 3rd, 2021

Prof. Mingming Zhou, Ph.D.; Academic Editor

RE: PONE-D-20-30355R1 - Can self-efficacy mediate between knowledge of policy, school support and teachers' attitudes towards inclusive education?

Dear Dr. Zhou,

We would like to thank you and the reviewers for your helpful feedback and comments on this manuscript. We have attempted to revise the manuscript accordingly. Please see our responses below to each comment.

We have reviewed our reference list and hope that it fits with journal guidelines.

Reviewer #1 did not have any additional comments.

Reviewer #2: Wanted us to further address the link between self efficacy and knowledge about inclusion policy, specifically to further address points 3 AND 5 of the previous comments. We added those previous comments here.

3. The link can be: attitudes → self efficacy → knowledge about policy →need for training module . may be this is a part of a bigger study but in the current form it requires a firm argument. A suggestion for clear organization is as under: 

The four aspects of inclusion in Israel

5. The three aspects on which the manuscript focuses are: self-efficacy, attitudes , knowledge about policy of inclusive education. Then why other variable such as school leadership , climate support system etc? 

We have reviewed these two comments. While we did feel that these two points are clarified in the manuscript, we did attempt to clarify them further. The two parts bellow were added to this revision. 

“School leadership has been consistently linked to teacher self-efficacy beliefs and attitudes [40, 41, 42] and has a direct relationship with school organizational climate [43, 44] and institutional support [24].”

“Put together, teacher beliefs in self-efficacy in implementing inclusionary practices have been found to be related to their attitudes regarding inclusion [46, 47]. Likewise, policy engagement and understanding is a clear aspect of organizational health and has been linked with leader self-efficacy [48]. Further, direct pre- and in-service training is clearly linked to perceived self-efficacy [49].”

We hope the editor and reviewers will find the revisions made to the manuscript to be satisfactory. We hope this manuscript, in its current format, will be found suitable for publication in Plos One. Thank you again for the opportunity to revise this article.

---

## [Decision Letter · Decision Letter 2]

8 Sep 2021

Can self-efficacy mediate between knowledge of policy, school support and teachers' attitudes towards inclusive education?

PONE-D-20-30355R2

Dear Dr. Werner,

We’re pleased to inform you that your manuscript has been judged scientifically suitable for publication and will be formally accepted for publication once it meets all outstanding technical requirements.

Kind regards,

Mingming Zhou, Ph.D.

Academic Editor

PLOS ONE

Additional Editor Comments (optional):

Reviewers' comments:

Reviewer's Responses to Questions

**Comments to the Author**

1. If the authors have adequately addressed your comments raised in a previous round of review and you feel that this manuscript is now acceptable for publication, you may indicate that here to bypass the “Comments to the Author” section, enter your conflict of interest statement in the “Confidential to Editor” section, and submit your "Accept" recommendation.

Reviewer #2: All comments have been addressed

Reviewer #3: All comments have been addressed

2. Is the manuscript technically sound, and do the data support the conclusions?

Reviewer #2: Yes

Reviewer #3: Yes

3. Has the statistical analysis been performed appropriately and rigorously? 

Reviewer #2: Yes

Reviewer #3: Yes

4. Have the authors made all data underlying the findings in their manuscript fully available?

Reviewer #2: (No Response)

Reviewer #3: Yes

5. Is the manuscript presented in an intelligible fashion and written in standard English?

Reviewer #2: (No Response)

Reviewer #3: Yes

6. Review Comments to the Author

Reviewer #2: Thank you for addressing the comments. The manuscript reads better in the current form. congratulations for the hard work and patience.

Reviewer #3: The article version meets the requirements. The revisions have improved the quality of the article.

7. PLOS authors have the option to publish the peer review history of their article (what does this mean?). If published, this will include your full peer review and any attached files.

Reviewer #2: **Yes: **Meenakshi Srivastava (PhD)

Reviewer #3: No

---

## [Editor Report · Acceptance letter]

10 Sep 2021

PONE-D-20-30355R2 

Can self-efficacy mediate between knowledge of policy, school support and teacher attitudes towards inclusive education? 

Dear Dr. Werner:

I'm pleased to inform you that your manuscript has been deemed suitable for publication in PLOS ONE. Congratulations! Your manuscript is now with our production department. 

Kind regards, 

on behalf of

Dr. Mingming Zhou 

Academic Editor

PLOS ONE